# Cardiovascular Healthcare Cost Savings Associated with Increased Whole Grains Consumption among Adults in the United States

**DOI:** 10.3390/nu12082323

**Published:** 2020-08-03

**Authors:** Mary M. Murphy, Jordana K. Schmier

**Affiliations:** 1Exponent, Inc., Center for Chemical Regulation and Food Safety, Washington, DC 20036, USA; 2Pharmerit—An Open Health Company, Real-World Evidence and Data Analytics Center of Excellence, Bethesda, MD 20814, USA; jschmier@pharmerit.com

**Keywords:** cardiovascular disease, costs and cost analysis, nutrition economics, public health, whole grains

## Abstract

Little is known about the potential health economic impact of increasing the proportion of total grains consumed as whole grains to align with Dietary Guidelines for Americans (DGA) recommendations. Health economic analysis estimating difference in costs developed using (1) relative risk (RR) estimates between whole grains consumption and outcomes of cardiovascular disease (CVD) and a selected component (coronary heart disease, CHD); (2) estimates of total and whole grains consumption among US adults; and (3) annual direct and indirect medical costs associated with CVD. Using reported RR estimates and assuming a linear relationship, risk reductions per serving of whole grains were calculated and cost savings were estimated from proportional reductions by health outcome. With a 4% reduction in CVD incidence per serving and a daily increase of 2.24 oz-eq of whole grains, one-year direct medical cost savings were estimated at US$21.9 billion (B) (range, US$5.5B to US$38.4B). With this same increase in whole grains and a 5% reduction in CHD incidence per serving, one-year direct medical cost savings were estimated at US$14.0B (US$8.4B to US$22.4B). A modest increase in whole grains of 0.25 oz-eq per day was associated with one-year CVD-related savings of $2.4B (US$0.6B to US$4.3B) and CHD-related savings of US$1.6B (US$0.9B to US$2.5B). Increasing whole grains consumption among US adults to align more closely with DGA recommendations has the potential for substantial healthcare cost savings.

## 1. Introduction

Cardiovascular disease (CVD) is a significant health burden globally and while the age-standardized prevalence of CVD has declined in the US and other high-income countries over the last decades, it remains a leading cause of morbidity and mortality [1]. In 2016, prevalence of CVD, excluding hypertension, in the US population 20 years and older was 9.0%, representing 24.3 million adults; including hypertension, prevalence of CVD increases to 121.5 million adults [2]. The economic burden associated with CVD is sizeable and medical costs alone are expected to exceed US$750 billion by 2035 [2].

Epidemiological and clinical evidence provide support for the importance of diet in maintaining health and reducing risk for chronic disease. For example, consumption of whole grains is associated with several health benefits, including reduced risk for CVD [3,4]. The 2015–2020 Dietary Guidelines for Americans (DGA) recommends consumption of at least half of total grains as whole grains [5]. For a 2000-calorie diet, the recommended amount of grains in the Healthy U.S.-Style Eating Pattern is 6 ounce-equivalents per day (oz-eq/day) with at least 3 oz-eq/day consumed as whole grains. On a given day, however, whole grains account for approximately 16% of total grains consumed among US adults [6], indicating that most adults fall far short of meeting the dietary recommendation. 

Previous scenario analyses have modeled changes in healthcare costs from beneficial shifts in dietary patterns and provide an indication of the potentially substantial economic impact associated with modifications to the population’s diet [7,8,9,10,11]. The purpose of this study is to estimate the impact on healthcare costs associated with a reduced incidence of total CVD based on modeling incremental increases in whole grains consumption as a proportion of total grains up to levels recommended in the DGA.

## 2. Materials and Methods 

A spreadsheet model was developed to quantify potential reductions in healthcare costs using inputs derived from (1) relative risk (RR) estimates from meta-analyses quantifying the association between whole grains consumption and cardiovascular health outcomes (incidence and incidence or mortality of CVD, and coronary heart disease (CHD)); (2) estimates of total grains and whole grains consumption among US adults; and (3) published data on annual direct medical and indirect costs associated with CVD and risk factors. The base case analysis applied direct medical costs to the reduction in incidence and a sensitivity analysis applied total costs (direct medical and indirect) to incidence or mortality reduction.

### 2.1. Model Inputs

#### 2.1.1. Risk Estimates of CVD Incidence and Whole Grains Consumption

A search of the published literature was conducted to identify RR estimates from recent meta-analyses quantifying the association between whole grains consumption and cardiovascular health outcomes. The PubMed search was conducted in April 2020, using a combination of key words and Medical Subject Headings (MeSH) terms, and limited to English language papers indexed since 2009. The citations were screened to identify meta-analyses conducted on prospective cohorts of healthy adults at risk for chronic disease that provided quantitative measures of the association between consumption of whole grains and incidence of total CVD and selected components (CHD and stroke). Two meta-analyses were identified as candidate sources for the RR estimates [3,12], while a third study presented RR estimates for CHD, stroke, and heart failure but not total CVD [13]. The meta-analysis by Aune et al. [3] was selected for use in the economic analysis based on the availability of estimates for both CVD and CHD incidence and a more conservative (i.e., smaller risk reduction) estimate for CHD based on adjustment for multiple demographic and clinical factors not included in the assessment by Reynolds et al. [12]. Given the lack of a significant association between whole grains consumption and stroke incidence, stroke was not evaluated as a separate outcome. Aune et al. [3] reported RR for CVD and CHD incidence and incidence or mortality reported per 90 g of whole grains. Using the reported RR and assuming a linear relationship, we calculated risk reductions per serving of whole grains (assuming a 30 g serving) for use in this analysis (Table 1). These values serve as inputs to the model.

#### 2.1.2. Whole Grains Consumption by US Adults

Consumption of whole grains in the US is estimated in units of oz-eq in nationwide food consumption surveys, which aligns with recommendations in the DGA. In this study, a 30 g serving of whole grains as reported in the observational studies was assumed to approximate each oz-eq of whole grains, where an oz-eq of grains corresponds to the amount of food containing 16 g of flour or 28.35 g for uncooked grains or cereals [5]. Based on dietary data collected in 2015 and 2016, the mean consumption of total grains by adults (20 years and older) in the US was 6.34 ounce equivalents (oz-eq)/day, with 0.93 oz-eq as whole grains and 5.41 oz-eq/day as refined grains [14], indicating that on average, Americans consume approximately one-third of the whole grains recommended in the DGA. Meeting dietary guidance of at least one-half of total grains as whole grains would require that at least 3.17 oz-eq of the current total grains consumption (6.34 oz-eq) be whole grains [5]. The average daily whole grains “gap” between current and recommended intakes for US adults therefore is 2.24 oz-eq. 

In one modeled scenario, an increase of 2.24 oz-eq whole grains was applied, thus eliminating the whole grains gap at the mean level of intake across US adults. Given current consumption levels, this best-case scenario is optimistic and may not be readily achievable, therefore additional scenarios were developed, reflecting increases of 0.25, 0.5, 1, or 2 oz-eq whole grains per day. These incremental increases reflect a range of shifts in mean consumption of whole grains among adults, including small and potentially more attainable changes.

#### 2.1.3. Annual Direct and Indirect Healthcare Costs Associated with Cardiovascular Disease

Annual direct medical costs and indirect costs for the selected health outcomes were based on data from the American Heart Association (AHA) [2] and inflated to 2019 US dollars using the inflation factor for medical care [15]. The AHA Statistical Update provides annual estimates of costs for cardiovascular diseases, including direct medical costs and indirect costs, specifically, lost productivity attributed to mortality.

### 2.2. Analysis

In the base case analysis, summary RR estimates and corresponding lower and upper 95% confidence intervals quantifying the association between consumption of a serving of whole grains and CVD incidence and CHD incidence were combined with increased whole grains consumption under the scenarios considered in this study (0.25, 0.5, 1, 2, or 2.24 oz-eq/day increases). The 4% reduction in CVD incidence per serving, derived from the Aune et al. meta-analysis [3] and shown in Table 1, was multiplied by the modeled change (increase) in the number of servings per day to calculate the estimated reduction in RR (e.g., 9% = 4% reduction in CVD incidence per servings times 2.24 servings). Total annual direct medical costs were reduced proportionally to reflect the change in risk of each health outcome. Using the same approach, a sensitivity analysis applied total costs (direct medical and indirect) to incidence or mortality risk reduction.

## 3. Results

Using risk reduction parameters derived from the literature (Table 1), the estimated annual direct medical cost savings from reduced risk of CVD was US$21.9B (range of US$5.5B to US$38.4B) assuming half of all grains consumed are whole grains (i.e., an increase of 2.24 oz-eq/day), as shown in Table 2. The estimated annual direct medical cost savings from reduced risk of CHD was US$14.0B (US$8.4B to US$22.4B) assuming half of all grains consumed are whole grains. Proportionally lower cost savings were estimated in models assuming smaller increases in whole grains consumption, with increased intake of 0.25 oz-eq/day estimated to save US$2.4B annually (US$0.6B to US$4.3B) from reduced risk of CVD.

In the sensitivity analysis, total costs (direct medical and indirect) were applied to estimates of reduction in risk of incidence or mortality (Table 3). Under these analyses, the estimated annual cost savings with an increase of 2.24 oz-eq/day in whole grains consumption was US$36.0B (US$9.0B to US$63.1B) from reduced risk of CVD and US$28.0B (US$16.8B to US$44.9B) from reduced risk of CHD.

## 4. Discussion

Findings from this study indicate that increased consumption of whole grains as a proportion of total grains to align with dietary recommendations may result in substantial healthcare cost savings due to reduced risk for cardiovascular diseases. More complex approaches to health economic modeling are available such as cohort models and microsimulation models [16,17]. A cost-of-illness [7,8] or cost-benefit [10] approach, or a scenario analysis [11] approach, similar to the current analysis are commonly used for public health nutrition models, and these approaches are particularly appropriate when there are limited data about consumption of a specific food, the lag between increased (or decreased) consumption and the health effect, and how other population risk factors may affect the change in incidence.

Loewen and colleagues estimated the economic burden (total direct health and indirect costs) of low whole grains consumption on cardiovascular outcomes including ischemic heart disease, ischemic stroke, and hemorrhagic stroke at CAD$2.6 billion per year in 2018 [18]. We are not aware of other models assessing healthcare savings from increased consumption of whole grains in the general population in the US, therefore the results from this analysis provide the first estimate of potential US cost savings when whole grain consumption is directly aligned with the DGA. Lee and colleagues’ model explored the cost-effectiveness of financial incentives associated with increased whole grains and other dietary improvements in Medicare and Medicaid-covered US beneficiaries [17]. A 2019 microsimulation by Jardim and colleagues estimated that 9.3% of cardiometabolic direct medical costs in the US ($7.5B) were attributable to suboptimal whole grain consumption [16].

Given the substantial savings estimated with even a modest increase in whole grains consumption of 0.25 oz-eq/day, this study, along with others, supports the importance of promoting beneficial shifts in dietary patterns. These data may be particularly useful when establishing goals for public health programs, in that even dietary changes below DGA targets may have health economic benefits.

The change in the proportion of grains consumed as whole grains necessary to meet DGA recommendations is not trivial, and increasing consumption of whole grains may be a challenge. For example, an analysis of whole grains purchases by participants in the Special Supplemental Nutrition Program for Women, Infants, and Children (WIC) following changes to the WIC package in 2009 show a monthly increase in whole grain purchases of 4.34 oz-eq, which corresponds to an increase from 28.4% to 34.5% of total grains as whole grains [19]. While the increased purchases of whole grains reported are encouraging, these results indicate that substantial increases in whole grains consumption may be difficult; availability and access are necessary but not sufficient to encourage large changes in consumption. It is particularly challenging given that there is not yet consistency in how whole grains are defined. The Whole Grains Stamp is used globally to provide consumers with information on percent and total whole grain content [20] and is useful in the absence of a labeling requirement. As the scientific community reaches consensus on definitions and communication mechanisms, it will enhance consumers’ ability to recognize specific amounts of whole grains, facilitating more informed replacement of refined grains with whole grains.

There are several important limitations to consider in interpreting findings as well as for designing future studies. First, there was no consensus in the literature on the definition of a whole grain food or the amount of whole grains that constitutes a ‘serving’, thus there are inconsistencies in how whole grains consumption is captured, a challenge that the research and manufacturing community is currently trying to solve. Second, the model assumes a linear relationship between increased consumption of whole grains and risk of CVD or CHD, where each unit increase in whole grains consumption proportionally reduces risk and in turn proportionally reduces cost. Evidence of non-linear associations between consumption of whole grains and risk of CVD and CHD was reported for the estimates used in this model [3]. For CHD, the steepest dose-response was observed with consumption of up to 90 g whole grains (i.e., ~3 servings) while further, though less steep, reductions were seen with consumption of up to 210 g whole grains. For CVD, there was evidence of a linear dose-response up to approximately 50 g (i.e., ~1.5 servings) of whole grains and more modest declines with consumption of up to 200 g whole grains. Consequently, use of the effect size per 90 g to calculate the incremental change per unit serving of whole grains (30 g, assumed to be equivalent to 1 oz-eq) or fractions of a serving may underestimate reduction in CVD risk in the range of modeled changes. The assumption of linearity remains a concern, as is the potential that the association between incidence and costs is also not linear, which could further confound the results. The existing literature does not allow for the prediction of changes in the distribution of severity of disease; for example, there may not only be fewer adults with CVD, but there could be a shift toward milder and thus less costly disease. Similarly, the model assumes a single level of whole grains consumption based on the population level intakes; for some adults, a small change in whole grains consumption could meet DGA recommendations while for others following a Healthy US-Style dietary pattern, an increase of 2.24 oz-eq/day may be insufficient to meet recommendations. Analysis of whole grains intake by age, sex, and ethnicity suggests there are differences in consumption that might be applied to future cost analyses [6]. As more of these relationships are documented and assumptions are replaced with evidence, microsimulation modeling on the impact of increases in whole grains consumption will become increasingly informative. The model in the current analysis does not address other potential health outcomes and related costs, either positive or negative, related to increased whole grains consumption in place of refined grains. The model also does not address potential cost to the consumer, though some data suggest a smaller price differential between whole grains and non-whole grains than between other healthier and traditional foods [21]. As additional data are collected and identified to minimize the uncertainty in this preliminary analysis, this model can be refined.

Findings from this modeling study indicate that increasing whole grains consumption among US adults, from incremental increases to more substantial increases to align more closely with the DGA recommendation, has the potential to provide substantial savings in healthcare costs.

## Figures and Tables

**Table 1 nutrients-12-02323-t001:** Input parameters for reduction in risk of cardiovascular disease and coronary heart disease with consumption of whole grains.

Outcome	# of Studies	Relative Risk (95% Confidence Interval) per 90 g; *I*^2^; p_heterogeneity_	Relative Risk (95% Confidence Interval) per 30 g Serving	Change in Risk
Base case analysis				
CVD incidence	2	0.87 (0.78, 0.97); 0%; 0.85	0.96 (0.93, 0.99)	Decrease
CHD incidence	5	0.84 (0.77, 0.92); 34%; 0.20	0.95 (0.92, 0.97)	Decrease
Sensitivity analysis				
CVD incidence or mortality	10	0.78 (0.73, 0.85); 40%; 0.09	0.93 (0.91, 0.95)	Decrease
CHD incidence or mortality	7	0.81 (0.75, 0.87); 9%; 0.36	0.94 (0.92, 0.96)	Decrease

Source: Aune et al. 2016 [3]. RR per 30 g was calculated from the reported values assuming a linear relationship; CVD = cardiovascular disease; CHD = coronary heart disease; *I*^2^ = statistic used to assess statistical heterogeneity between studies; heterogeneity <60% is generally regarded as little to moderate; p_heterogeneity_ = a measure of statistical heterogeneity, where *p* < 0.10 is considered to be significant.

**Table 2 nutrients-12-02323-t002:** Annual direct medical cost savings associated with increased consumption of whole grains among adults in the US.

Model Scenarios: Increase in Whole Grains (Ounce-Equivalent/Day)	Annual Direct Medical Cost Savings (US$ Billions) ^a^ Direct (Lower–Upper Range) Based on Relative Risk of Disease Incidence	
	Cardiovascular Disease	Coronary Heart Disease
2.24	21.9 (5.5–38.4)	14.0 (8.4–22.4)
2.0	19.6 (4.9–34.3)	12.5 (7.5–20)
1.0	9.8 (2.4–17.1)	6.3 (3.8–10)
0.5	4.9 (1.2–8.6)	3.1 (1.9–5)
0.25	2.4 (0.6–4.3)	1.6 (0.9–2.5)

^a^ Annual direct medical costs (inflated to 2019 US dollars): Cardiovascular disease costs = US$244.8 (Billions, B), where outcomes include heart disease, hypertension, stroke, and other circulatory conditions. Coronary heart disease costs = US$125.3 (Billions, B), where outcomes include CHD, heart failure, part of hypertension, cardiac dysrhythmias, rheumatic heart disease, cardiomyopathy, pulmonary heart disease, and other or ill-defined heart disease [2,15]; RR normalized to effect per 30 g serving of whole grains assuming linear response [3].

**Table 3 nutrients-12-02323-t003:** Sensitivity analysis: Net direct medical and indirect cost savings associated with increased consumption of whole grains among adults in the US.

Model Scenarios: Increase in Whole Grains (Ounce-Equivalent/Day)	Annual Cost Savings (US$ Billions) ^a^ Total (Lower–Upper Range) Based on Relative Risk of Disease Incidence or Mortality	
	Cardiovascular Disease	Coronary Heart Disease
2.24	36.0 (9.0–63.1)	28.0 (16.8–44.9)
0.25	4.0 (1.0–7.0)	3.1 (1.9–5.0)

^a^ Annual average costs (inflated to 2019 US dollars): Cardiovascular disease costs = US$244.8 (billions, B) direct costs, US$157.4B indirect costs, and US$402.2B total costs, where outcomes include heart disease, hypertension, stroke, and other circulatory conditions. Coronary heart disease costs = US$125.3 (billions, B) direct costs, US$125.1B indirect costs, and US$250.4B total costs, where outcomes include CHD, heart failure, part of hypertension, cardiac dysrhythmias, rheumatic heart disease, cardiomyopathy, pulmonary heart disease, and other or ill-defined heart disease [2,15]; RR normalized to effect per 30 g serving of whole grains assuming linear response [3].

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
