# Peer review of "Cardiovascular Healthcare Cost Savings Associated with Increased Whole Grains Consumption among Adults in the United States"

_nutrients, 2020, doi:10.3390/nu12082323_

Round 1

Reviewer 1 Report

Summary of Paper

 The objective of this paper is to estimate the impact on healthcare costs associated with a reduction in the incidence and mortality from cardiovascular disease (CVD) and chronic heart disease (CHD) emanating from an increase in the consumption of whole grains as a proportion of total grains. The studied change in proportion would bring the proportion more in line with the U.S. recommended dietary guidelines.

The authors take a simple cost-of-illness approach with inputs to the model obtained from secondary data sources to develop their estimates. They find that increasing whole grain consumption as a proportion of total grain consumption among US adults to align more closely with U.S. dietary guidelines has the potential to reduce healthcare costs associated with CVD and CHD.

Assessment of Paper

The paper is clearly written and the approach is in line with similar studies. In preparing this report, I carefully read some similar papers cited the manuscript (references 7-11). I have some suggestions for minor revisions to improve the readability of the paper and some methodological comments that should be addressed.  

Comments    

  1. Abstract: I think you should rename “health economic model” to “cost-of -illness” analysis because this is what it really is. A health economic model implies a behavioral model grounded in economics. This is not what you have. Your analytical approach is similar to references 7, 8, and 11. References 7 and 8 published in Nutrients call their analysis a cost-of-illness analysis.
  1. An important challenge in estimating net change in cost associated with some type of intervention such as the one you are considering in this paper on chronic health conditions is over-estimating the cost savings associated with a reduction in the incidence of disease because of co-morbidities. Scrafford et al 2019 (ref. 11 in paper) address this in their paper on p. 607. Even though you are not considering type 2 diabetes (T2D), it is an established risk factor for heart disease and the risk for developing HD or T2D are both affected by dietary fiber intake. Some of the cost savings estimated for HD in this paper are likely overstated because they are really attributable to costs associated with T2D in people with both conditions.

At a minimum, this possibility should be addressed as a limitation of your analysis.

  1. page 2, line 51: I suggest changing “deterministic model” to “cost-of-illness analysis”. You do not have a deterministic model, you have at best an associative model. Deterministic implies causal. The analytical approach taken in this paper cannot be considered causal.

  1. page 2, line 54: I would eliminate stroke as a health outcome since you do not consider it.

  1. page 2, line 74: I suggest changing “economic model” to “cost-of-illness analysis”.

  1. page 2, .lines 75-78: The writing here is hard to follow. Maybe break it up into more than one sentence.

  1. Page 2, line 76 and Table 1. I do not understand what is meant by CVD incidence/mortality as it is written in the paper. I had to go to Aune et al to figure this out. At first glance, I thought you had some measure of incidence rate divided by mortality rate which did not make a lot of sense. But what you have is some combined risk reduction from either incidence of CVD or CHD and mortality from CVD or CHD. Simply saying incidence or mortality instead of incidence/mortality with make this clearer.

  1. page 3, lines 84-99: You introduce the term oz-eq/day without saying what it is until the end of the paragraph. I suggest introducing this sooner in the paragraph and explaining why you are choosing this metric instead of sticking with grams.

  1. page 3, lines 100-105: Another area contributing to an overestimation of potential cost savings is assuming a one to one reduction in healthcare and indirect costs associated with a reduction in incidence of CHD or CVD. Abdullah et al 2017 (ref 7 in this paper) address this in their analysis by adjusting the reduction in hospital costs for fixed and variable costs. I would argue that outpatient costs should similarly account for fixed and variable costs.

  1. page 4, Table 3: Why do treat the incidence or mortality analysis as a sensitivity analysis? Why is not part of the main analysis?

  1. page 4, line 153: I suggest differentiating the approaches taken in the cited studies in ref. 7-11. Ref. 7, 8, and 11 take a cost-of-illness approach which is most closely related to your approach. Ref. 9 takes a microsimulation approach and ref 10 takes a cost-benefit analysis approach.
  2. Page 5, line 196: Another important point for discussion that fits in with this paragraph is the fact that not everybody responds well to carbohydrates of any kind. Some people do not process carbohydrates well; therefore the dietary guidelines studied in this paper may not be beneficial to such people.

Author Response

REVIEWER 1: Comments and our responses (provided in red font)

Comments and Suggestions for Authors

Summary of Paper

The objective of this paper is to estimate the impact on healthcare costs associated with a reduction in the incidence and mortality from cardiovascular disease (CVD) and chronic heart disease (CHD) emanating from an increase in the consumption of whole grains as a proportion of total grains. The studied change in proportion would bring the proportion more in line with the U.S. recommended dietary guidelines.

The authors take a simple cost-of-illness approach with inputs to the model obtained from secondary data sources to develop their estimates. They find that increasing whole grain consumption as a proportion of total grain consumption among US adults to align more closely with U.S. dietary guidelines has the potential to reduce healthcare costs associated with CVD and CHD.

Assessment of Paper

The paper is clearly written and the approach is in line with similar studies. In preparing this report, I carefully read some similar papers cited the manuscript (references 7-11). I have some suggestions for minor revisions to improve the readability of the paper and some methodological comments that should be addressed.

Comments

  1. Abstract: I think you should rename “health economic model” to “cost-of -illness” analysis because this is what it really is. A health economic model implies a behavioral model grounded in economics. This is not what you have. Your analytical approach is similar to references 7, 8, and 11. References 7 and 8 published in Nutrients call their analysis a cost-of-illness analysis.

Response: We appreciate your careful review of our paper. We note that there are many types of health economic analyses and a cost-of-illness approach is one type. We have used cost-of-illness data from AHA as one input in our analysis to estimate potential cost savings, though our analysis is not a cost-of-illness analysis per se.1 Based on your suggestion, we have changed “health economic model” to “health economic analysis” to better reflect the approach we’ve taken. As noted below in our response to item 3, we have referred to it as a spreadsheet analysis. Lines 14, 56

1 https://www.nlm.nih.gov/nichsr/hta101/ta10107.html

2. An important challenge in estimating net change in cost associated with some type of intervention such as the one you are considering in this paper on chronic health conditions is over-estimating the cost savings associated with a reduction in the incidence of disease because of co-morbidities. Scrafford et al 2019 (ref. 11 in paper) address this in their paper on p. 607. Even though you are not considering type 2 diabetes (T2D), it is an established risk factor for heart disease and the risk for developing HD or T2D are both affected by dietary fiber intake. Some of the cost savings estimated for HD in this paper are likely overstated because they are really attributable to costs associated with T2D in people with both conditions.
At a minimum, this possibility should be addressed as a limitation of your analysis.

Response: We understand the importance of the concept of overstating costs and agree this was a necessary consideration in the analysis by Scrafford et al., (2019) in which costs from different outcomes were summed. In the current analysis, we are not adding costs savings for CVD with costs savings for T2D (or other conditions). The meta-analysis upon which we rely for our risk reduction estimates did not exclude patients with diabetes, therefore the CVD reductions can be assumed to be relevant for all populations (both with and without T2D).

3. page 2, line 51: I suggest changing “deterministic model” to “cost-of-illness analysis”. You do not have a deterministic model, you have at best an associative model. Deterministic implies causal. The analytical approach taken in this paper cannot be considered causal.

Response: We have changed the wording to a spreadsheet model to more accurately describe our approach. Lines 56

4. page 2, line 54: I would eliminate stroke as a health outcome since you do not consider it.

Response: We eliminated stroke as a health outcome, though we maintained text in our methods to explain how it was excluded from the analysis. Lines 59, 72-79

5. page 2, line 74: I suggest changing “economic model” to “cost-of-illness analysis”.

Response: We deleted this portion of the sentence to streamline information on stroke as an outcome. Lines 79

6. page 2, .lines 75-78: The writing here is hard to follow. Maybe break it up into more than one sentence.

Response: Thank you for this feedback. We revised the sentence as suggested. Lines 79-85

7. Page 2, line 76 and Table 1. I do not understand what is meant by CVD incidence/mortality as it is written in the paper. I had to go to Aune et al to figure this out. At first glance, I thought you had some measure of incidence rate divided by mortality rate which did not make a lot of sense. But what you have is some combined risk reduction from either incidence of CVD or CHD and mortality from CVD or CHD. Simply saying incidence or mortality instead of incidence/mortality with make this clearer.

Response: Thank you for this feedback. This change was made throughout the manuscript. Lines 59, 63, 83, Table 1, 130, 149, Table 2

8. page 3, lines 84-99: You introduce the term oz-eq/day without saying what it is until the end of the paragraph. I suggest introducing this sooner in the paragraph and explaining why you are choosing this metric instead of sticking with grams.

Response: We added a sentence to provide rationale for using the term oz-eq and moved the definition closer to the beginning of the paragraph. Lines 94-98

9. page 3, lines 100-105: Another area contributing to an overestimation of potential cost savings is assuming a one to one reduction in healthcare and indirect costs associated with a reduction in incidence of CHD or CVD. Abdullah et al 2017 (ref 7 in this paper) address this in their analysis by adjusting the reduction in hospital costs for fixed and variable costs. I would argue that outpatient costs should similarly account for fixed and variable costs.

Response: We appreciate that addressing fixed and variable costs was appropriate for an assessment in Canada, though in the US healthcare system, fixed and variable costs are not explicitly taken into account as part of the diagnosis reimbursement system and applying an adjustment factor would not be appropriate.

10. page 4, Table 3: Why do treat the incidence or mortality analysis as a sensitivity analysis? Why is not part of the main analysis?

Response: The incidence or mortality analysis was treated as a sensitivity analysis to limit the primary analysis to more conservative cost estimates.

11. page 4, line 153: I suggest differentiating the approaches taken in the cited studies in ref. 7-11. Ref. 7, 8, and 11 take a cost-of-illness approach which is most closely related to your approach. Ref. 9 takes a microsimulation approach and ref 10 takes a cost-benefit analysis approach.

Response: Thank you for this suggestion. The references were separated to more clearly link each citation to an approach Lines 166-168

12. Page 5, line 196: Another important point for discussion that fits in with this paragraph is the fact that not everybody responds well to carbohydrates of any kind. Some people do not process carbohydrates well; therefore the dietary guidelines studied in this paper may not be beneficial to such people.

Response: Our intent with this analysis is to model costs with population level increases in the consumption of whole grains as a proportion of total grains. We appreciate that some individuals may follow dietary patterns that deviate from patterns consistent with the DGA for health or other reasons. We added a note to indicate that the recommendations are for adults following a US Healthy-Style dietary pattern. Lines 221-222

Submission Date
22 June 2020
Date of this review
24 Jul 2020 22:11:42

Reviewer 2 Report

-line 33/34 are there more recent data than 2016 you can cite? Same comment for line 85.

-line 46/47 suggest incorporating the new paragraph into the previous one rather than have a very short paragraph

-line 50-58: a lot of information in this paragraph. I hope each part of your model and methodology is explained carefully in the pages that follow

-Table 1, columns 2 and 3. Column titles need to be clear so the reader knows what is happening. Spell out rather than shorthand (i.e. "CI"). Also be sure to explain in the supporting text what the statistics mean.

-overall good work.

Author Response

REVIEWER 2: Comments and our responses (provided in red font)

Comments and Suggestions for Authors

-line 33/34 are there more recent data than 2016 you can cite? Same comment for line 85.

Response: We are aware that updated prevalence data for some conditions have been released with NHANES 2017-18 data (e.g., hypertension); we are not aware of updated estimates of CVD prevalence as presented in the AHA reference at this time.

-line 46/47 suggest incorporating the new paragraph into the previous one rather than have a very short paragraph

Response: The paragraphs were combined. Line 51

-line 50-58: a lot of information in this paragraph. I hope each part of your model and methodology is explained carefully in the pages that follow

Response: All input parameters and the methods used to combine the data are detailed in Lines 65-130.

(1) relative risk (RR) estimates from meta-analyses quantifying the association between whole grains consumption and cardiovascular health outcomes (incidence and incidence or mortality of CVD, and coronary heart disease (CHD)): detailed in Lines 65-92, Table 1

(2) estimates of total grains and whole grains consumption among US adults: detailed in Lines 93-113

(3) published data on annual direct medical and indirect costs associated with CVD and risk factors: detailed in Lines 114-119; Tables 2 and 3

Methods: detailed in Lines 120-130

-Table 1, columns 2 and 3. Column titles need to be clear so the reader knows what is happening. Spell out rather than shorthand (i.e. "CI"). Also be sure to explain in the supporting text what the statistics mean.

Response: We have spelled out CI as suggested (and RR) and included definitions for additional statistics in the footnotes.

-overall good work.

Response: Thank you for this feedback

Submission Date
22 June 2020
Date of this review
27 Jul 2020 14:20:31

Reviewer 3 Report

Abstract: I suggest summarizing the analysis on Page 3, Lines 85-90.

-Contextualizing the 2.24 oz-eq figure or "whole Grains gap" would help a reader understand why a value of 2.24 oz-eq
is particularly important.

Page 1, Lines 40-41:It is helpful to give the complete DGA recommendation.

- The Dietary Guidelines recommend 3 or more ounce-equivalents of whole grain products per day (based on a 2,000 kcal diet),  and that at least half of all grains consumed be whole grains.
- Then on page 3, Line 86, you may consider referring back to this by saying that Americans -- who I assume are taking in  around 2,000 kcal per day -- are only about 1/3 of the way to the DGA recommendation. 

Page 3, Lines 94-96: The authors recognize that the "whole grains gap" is large and may not easily be filled.

-Direct support of this statement would be helpful. For example, Oh, Jensen, and Rahkovsky (2016)[https://doi.org/10.1093/aepp/ppw020] evaluated the impact of WIC package changes that were mandated by U.S. state agencies. The WIC program changes, which were required by October 2009, among other things, included more whole grain products in state agency WIC food lists.
-The authors show in Table 5 that, among WIC participants, the ounces of whole grains in purchases went up from 44.31 to 48.65 (or 4.34 ounces). As percentage of total grains, this was an increase from 28.4% to 34.5%. Of course, not all of the increase was due to the package change -- but even assuming that it is, such a major program change targeting the WIC population resulted in a modest increase in WIC purchases. I think including a discussion of their finding can help the reader understand how difficult changing whole grains consumption may be, especially to get Americans to the DGA mark of 50% of total grains being whole grains.

Author Response

REVIEWER 3: Comments and our responses (provided in red font)

Comments and Suggestions for Authors

Abstract: I suggest summarizing the analysis on Page 3, Lines 85-90.
Response: Text was added to the abstract on the method of analysis. Lines 18-20

-Contextualizing the 2.24 oz-eq figure or "whole Grains gap" would help a reader understand why a value of 2.24 oz-eq is particularly important.

Page 1, Lines 40-41:It is helpful to give the complete DGA recommendation.

-The Dietary Guidelines recommend 3 or more ounce-equivalents of whole grain products per day (based on a 2,000 kcal diet), and that at least half of all grains consumed be whole grains.

-Then on page 3, Line 86, you may consider referring back to this by saying that Americans -- who I assume are taking in around 2,000 kcal per day -- are only about 1/3 of the way to the DGA recommendation.

Response: Thank you for these suggestions; they serve to contextualize the 2.24 oz-eq figure and add to the clarity of our paper. The changes were made. Lines 94-98; 100-102,105-107

Page 3, Lines 94-96: The authors recognize that the "whole grains gap" is large and may not easily be filled.
-Direct support of this statement would be helpful. For example, Oh, Jensen, and Rahkovsky (2016)[https://doi.org/10.1093/aepp/ppw020] evaluated the impact of WIC package changes that were mandated by U.S. state agencies. The WIC program changes, which were required by October 2009, among other things, included more whole grain products in state agency WIC food lists.

-The authors show in Table 5 that, among WIC participants, the ounces of whole grains in purchases went up from 44.31 to 48.65 (or 4.34 ounces). As percentage of total grains, this was an increase from 28.4% to 34.5%. Of course, not all of the increase was due to the package change -- but even assuming that it is, such a major program change targeting the WIC population resulted in a modest increase in WIC purchases. I think including a discussion of their finding can help the reader understand how difficult changing whole grains consumption may be, especially to get Americans to the DGA mark of 50% of total grains being whole grains.

Response: Thank you for identifying this paper. This is an interesting study and we have incorporated it into our draft to illustrate the challenges in meeting the gap. Lines 188-194

Submission Date
22 June 2020
Date of this review
30 Jun 2020 10:45:30